# Adjuvants to improve efficacy of miticides in managed honey bee (*Apis mellifera*) colonies to control *Varroa destructor*

**Brandon Shannon**[ID]**[1]\*, Rui Zhang[2], Lucas Marsh[1], Reed M. Johnson**[ID]**[1]**

1 Department of Entomology, The Ohio State University, Wooster, Ohio, United States of America,
2 Department of Statistics, The Ohio State University, Columbus, Ohio, United States of America

\* Shannon.325@OSU.edu

## Abstract

Beekeepers must manage *Varroa destructor* mites to maintain colony health. Large-scale beekeepers often use chemical treatments (miticides) to manage this pest. Miticide resistance drives a need for compounds with alternative modes of toxic action that can be used in rotation as part of a *Varroa* management plan. This research aimed to determine the efficacy of oxalic acid, clove oil, and fenpyroximate when delivered in glycerin soaked in strips and combined with a range of bee-safe adjuvants. Adjuvants are a group of compounds used in plant pesticide applications to increase the spreading and penetration of a pesticide. Laboratory cage trials tested a miticidal active ingredient (oxalic acid, clove oil, or fenpyroximate) and an adjuvant (Ecostep BC-12°, Ecostep SE-11°, Ecostep AE-13°, Ecostep CE-13°, or Silwet L-7500°) in glycerin-soaked strips. Field trials evaluated the best performing active ingredient-adjuvant combination from cage trials, oxalic acid combined with Ecostep BC-12° adjuvant in glycerin-soaked strips. Neither glycerin control, oxalic acid alone, or oxalic acid with adjuvant caused a significant change in *Varroa* per 100 bees in field trial year 1, when starting *Varroa* levels were high (average 11.8 *Varroa* per 100 bees across all treatment groups). In year 2, when starting *Varroa* levels were low (average 0.58 *Varroa* per 100 bees across all treatment groups), *Varroa* per 100 bees increased 2.6-fold for the glycerin control and 2.8-fold for oxalic acid alone, while a 29% reduction was observed in the oxalic acid with adjuvant treatment. Additionally, mite drop data indicated increased speed for the miticidal effect when an adjuvant is included with oxalic acid. This research informs formulation chemistries for oxalic acid and other miticides to help beekeepers maintain healthy hives.

## Introduction

Honey bees (*Apis mellifera*) are responsible for pollination of at least 100 crops [1–3], with an estimated contribution of 12 billion dollars to the US economy [4,5] and 153

**Data availability statement:** All relevant data are within the manuscript and its Supporting Information files.

**Funding:** We thank the National Honey Board with Project Apis m., the California State Beekeeper's Association, The Ohio State University CFAES Internal Grants Program, and state and federal appropriations to The Ohio State University College of Food, Agriculture and Environmental Science (OHO01558-MRF). This work was supported by the Specialty Crop Research Initiative, project award no. 2023-51181-41246, from the U.S. Department of Agriculture's National Institute of Food and Agriculture. Any opinions, findings, conclusions, or recommendations expressed in this publication are those of the authors and should not be construed to represent any official USDA or U.S. Government determination or policy. The funders had no role in study design, data collection and analysis, decision to publish, or preparation of the manuscript.

**Competing interests:** A patent application (PCT/US2024/056156; ADJUVANTS TO IMPROVE EFFICACY OF VARROA CONTROL ACTIVE INGREDIENTS IN MANAGED HONEY BEE COLONIES) was filed on November 15, 2024, by the Ohio State Innovation Foundation with inventors Reed Johnson and Brandon Shannon. This does not alter our adherence to PLOS ONE policies on sharing data and materials.

billion USD worldwide [6]. However, high colony losses have been reported by commercial beekeepers, at an annual colony loss rate of 55% and winter loss rate of 37% in the United States [7] and a winter loss rate of 34% in Canada in 2023–2024 [8]. Parasitism by the ectoparasitic mite *Varroa destructor* is a major driver of colony loss. In surveys, 60% of beekeepers self-reported that *Varroa* and its associated diseases were the leading cause of winter colony losses in U.S. commercial beekeeping operations [9,10].

Because many of the non-chemical management strategies for *Varroa* are not sufficiently effective or too labor-intensive [11,12], large-scale beekeepers must use chemical treatments, known as miticides, to manage this pest. However, over-reliance and misuse of synthetic miticides have led *Varroa* to develop worldwide resistance to the formamidine amitraz [13–18], the organophosphate coumaphos [19–22], and the pyrethroids fluvalinate and flumethrin [23–26]. Because of widespread miticide resistance, there is a rising need for products with alternative modes of toxic action that can be used in rotation as part of a *Varroa* management plan. Potential alternatives explored by this study include extended-release formulations of the natural pesticides, oxalic acid and clove oil, and the synthetic pyrazole, fenpyroximate.

Oxalic acid is a miticide that is permitted for use in the United States, Canada, several European countries, and New Zealand [12], and has been used to control *Varroa* for several decades [27]. The mechanism of toxic action of oxalic acid to *Varroa* is not well understood [28], though effects of physical damage to the chitin plate of *Varroa* have been observed [29]. However, oxalic acid is not effective against *Varroa* protected by brood cell cappings, so it has limited efficacy when applied as a vaporization or dribble flash treatment when brood is present [30,31]. At the time of this study, no oxalic acid extended-release treatments were available for use in the United States, but many beekeepers treated with off-label oxalic acid extended release formulations using sponges or shop towels soaked in a 1:1 w/w mixture of glycerin and oxalic acid [32]. While clove oil (principal constituent eugenol) does not have a defined mode of action, it has demonstrated contact toxicity to *Varroa* [33–36]. Clove oil can act as an insect repellent and prohibits the growth of bacteria and fungi [37], but likely affects *Varroa* through effects on the enzymes glutathione-S-transferase, superoxide dismutase, and $Ca^{2+}$-$Mg^{2+}$-ATPase [38]. Fenpyroximate is a synthetic pyrazole acaricide in IRAC class 21A Mitochondrial Complex I Electron Transport Inhibitor [39] that was previously registered for in-hive use as Hivastan® and demonstrates promising mite control in semi-field evaluations [40]. Oxalic acid is registered for use as a dribble or vaporization treatment as Api-Bioxal®, as a vaporization treatment as EZ Ox®, or as an extended-release strip formulation as Varroxsan® [41], though only Api-Bioxal® was registered for use at the start of this experiment. Neither clove oil nor fenpyroximate are currently formulated in registered *Varroa* control products.

Adjuvants are added to pesticides, either as formulation components or tank-mix components, to improve the handling or application characteristics and enhance pesticide activity [42]. While adjuvants are typically used with spray applications in agriculture, they may also be included in some pesticides used to control common bee pests in managed honey bee colonies [43], though the identity of adjuvant ingredients

in these products are proprietary. The "principal functioning agents" that provide the desired function of an adjuvant are drawn from the list of inert ingredients maintained by US EPA and consist of the same or similar compounds used as formulation components in traditional pesticide products [44]. While some adjuvants have shown high toxicity to honey bees, others are relatively non-toxic [43,45]. Adjuvants are widely used and were valued globally at 3.8 billion USD in 2023 [46], but we could find no published studies that sought to make use of bee-safe adjuvants to improve pesticide efficacy within honey bee colonies. Using adjuvants that demonstrate low toxicity to bees as formulation components in *Varroa* control applications can potentially improve miticide activity and increase the number of active ingredients available for control of *Varroa* mites [47].

Oxalic acid dihydrate, clove oil, and fenpyroximate have shown promise as miticides for *Varroa* control, but demonstrate limited efficacy at concentrations that are safe for honey bees. This research aimed to determine the efficacy of these active ingredients when delivered in glycerin soaked strips and combined with a range of bee-safe adjuvants. Combinations of active ingredients and adjuvant were first tested in laboratory cage trials to identify promising miticide-adjuvant combinations. The most promising combination, oxalic acid active ingredient with Ecostep BC-12 adjuvant, was then assessed in field trials to determine if the addition of an adjuvant could improve *Varroa* control in whole honey bee colonies.

## Materials and methods

### Miticides and adjuvants

The miticides used in the laboratory cage trial included clove oil (100%; Sigma-Aldrich; MO, USA); fenpyroximate (>98.0%; Alfa Chemistry, Inc.; NY, USA); and oxalic acid dihydrate (99.5–102.5%; Thermo Fisher Sceintific; MA, USA). The miticide used in the field trial was oxalic acid dihydrate.

The adjuvants used in the laboratory cage trials are listed in Table 1 and include Ecostep AE-13® (Stepan, Northfield, IL, USA), Ecostep BC-12® (Stepan), Ecostep CE-13® (Stepan), Ecostep SE-11® (Stepan), and Silwet L-7500 Copolymer® (Momentive Performance Materials, Niskayuna, NY, USA). All four Ecostep products are listed for organic use through the Organic Materials Review Institute (OMRI). The principal functioning agents for all adjuvants are included in the list of inert ingredients safe for food and nonfood use [44]. The adjuvant used in the field trial was Ecostep BC-12®.

### Adjuvant acute toxicity tests

All adjuvants were tested for contact bee toxicity using a Potter Spray Tower [48]. The insecticide Mustang Maxx® (9.15% zeta-cypermethrin active ingredient), a pyrethroid insecticide with high acute toxicity to honey bees, was used as a positive control [45,49–51]. The 48-hour acute contact $LC_{50}$ of all adjuvants were determined using 3-day-old adult worker bees following methods described in [45]. Data analysis was performed with the *drc* package in R [52] using 2-parameter log-logistic dose-response models, as performed by Shannon et al. [45]. The concentrations tested were as follows: for the four Ecostep® products, 0, 1, 3, 5, 10, and 20% by volume; for Silwet L-7500 Copolymer®, 0, 0.3, 1, 3, 5, 10, and 20%

**Table 1. Adjuvant treatments used in this experiment. An asterisk (\*) indicates that the concentration of principal functioning agents is not given and the SDS states that the components are not hazardous or are below required disclosure limits.**

| Treatment | Manufacturer | Principal Functioning Agents (% Composition) |
|---|---|---|
| Ecostep AE-13® | Stepan | Polyethylene Glycol/ Polypropylene Glycol – (n/n) Copolymer (> 99%) |
| Ecostep BC-12® | Stepan | Polyethylene/ Polypropylene Glycol Monobutyl Ether (90–100%) |
| Ecostep CE-13® | Stepan | Castor Oil, Ethoxylated (90–100%) |
| Ecostep SE-11® | Stepan | Sorbitan, Mono-(9z)-octadecenoate, Poly(oxy-1,2-ethanediyl) Derivs (90–100%) |
| Silwet L-7500 Copolymer® | Momentive Performance Materials | Polyalkyleneoxide modified polydimethylsiloxane* |

by volume; for Mustang Maxx® Positive Control, 0.00186, 0.00558, 0.0186, 0.0558, and 0.186% by volume of formulation, which is 1.70e-4, 5.10e-4, 1.70e-3, 5.10e-3, 1.70e-2% zeta-cypermethrin active ingredient by volume [45,51].

## Cage trials

**Treatment preparation.**  Treatment solutions were prepared by adding glycerin (Fisher Scientific; MA, USA) to a 15 mL conical tube (Thermo Fisher Scientific; MA, USA) and sonicating with heating (Kendal Digital Ultrasonic Heated Cleaner model HB-S-49DHT) for 30 min to make 3 g of final treatment mixture. Negative control treatment solutions containing only glycerin were also heated. Clove oil treatments were prepared by adding clove oil to form a 3% v/w mixture with heating at 34º C. Fenpyroximate treatments were prepared by first dissolving solid fenpyroximate in acetone to a 10% w/v solution, then adding the fenpyroximate-acetone solution to glycerin to produce a 0.2% w/w solution that was heated to 34º C. Oxalic acid treatments were prepared by adding oxalic acid dihydrate to glycerin, up to, but not above, 67º C [53–56] and sonication for a minimum of 30 minutes, until crystals were no longer visible, to produce a 20% w/w solution. The concentrations of the active ingredients in cage trials were chosen based on preliminary studies that determined the maximum concentrations that could be applied with minimal mortality observed in caged bees.

Active ingredient with adjuvant combination treatments were prepared by adding a single adjuvant product (Ecostep AE-13˚, Ecostep BC-12˚, Ecostep CE-13˚, Ecostep SE-11˚, or Silwet L-7500 Copolymer˚) to the glycerin-active ingredient mixture and mixed fully to produce a 0.5% w/w solution for the Ecostep products or a 0.2% w/w solution for Silwet L-7500 Copolymer. The final active ingredient concentration was the same as in the active ingredient controls. The concentration of adjuvant was chosen based on the labeled concentrations for similar adjuvants in agricultural tank mix applications, which typically range from 0.0625 to 0.625 percent by volume [45].

Cage treatment strips were made from a Swedish sponge (Superscandi Swedish Dishcloths; London, England) cut into 32 sections (1.25 x 8.5 cm). A single treatment strip was placed in each 3 g solution and heated with sonication at the respective temperature for each treatment for 30 minutes. The concentration of 1/32 saturated treatment strip was used in cages to scale down a full-size strip that would be used with 10,000 bees in a deep Langstroth box [57,58] to approximately 300 bees in each cage.

**Cage design.**  The design of cages above petroleum jelly-coated weigh boats was modified from Rinkevich, 2020 [17] (S1 Text, Fig A). Modifications include the following: a 15 mm x 3 mm slit cut in the top of the cage to insert the 1.25 x 8.5 cm treatment strip; four equally spaced 3-mm holes were added for airflow on the sides of the cage, 1 cm from the top; two 3–5 g sucrose cubes were fixed to the top of the cage using hot melt adhesive to allow bees to feed.

**Honey bees.**  To populate the cages, honey bees were shaken from brood frames from six colonies, in July through October in 2022 and 2023. Colonies were managed at The Ohio State University – Wooster campus apiaries and had not been treated for mites for at least 6 months prior to collection. All colonies had been requeened with New World Carniolan queens (Olivarez Honey Bees, Orland, CA) in the Spring, but queens were caged for at least 21 days prior to worker bee collection to maximize the number of mites in the dispersal phase. Bees were shaken into an empty 4-frame nucleus box and stored in darkness for no longer than one hour before transferring to cages. Bees were misted with DI water during transfer to discourage flight. Approximately 300 bees were scooped from the nucleus box and placed into each cage containing a treatment.

**Experimental design.**  Each cage treatment series consisted of 6 treatments: a negative control (glycerin solvent only), active ingredient control (active ingredient in glycerin), and four combinations of that same active ingredient in glycerin with four different adjuvants. Using a cage assay rather than a traditional topical application or glass vial application better simulates the exposure that mites would face in a colony environment, but may introduce additional sources of variability compared to other topical application methods [59], from variable age [60–62], *Varroa* loads [59], bee health [12,63,64], and colony strength [65]. This variability was managed by performing each set of treatments, assigned in random order, with the same cohort of bees. A minimum of 3 replicates were performed for each treatment. Cages were

placed inside an incubator (Humidaire Model No. 2048; The Humidaire Incubator Company, New Madison, OH, USA) and stored at hive conditions (34° C, 60% humidity, darkness). Dead bees at the bottom of the cage and dead *Varroa* on the collection tray were counted after 24 hours. After bees and *Varroa* were counted, the plastic tray was discarded and cages were inverted and frozen at −20° C.

Bees were weighed (Ohaus, Parsippany, NJ; Model CL5000) and the number of grams was multiplied by 11.34 to estimate the number of bees, as determined in preliminary trials. Bees were agitated for 30 minutes in 70% ethanol to dislodge *Varroa* remaining on the bees for counting [66,67]. Recovery efficacy of alcohol washes was not evaluated, but the same methods were used for all samples. Treatment efficacy was determined by dividing the number of *Varroa* that had fallen during treatment by the total number of *Varroa* (mites fallen plus mites in alcohol wash). A logistic regression was used to determine significant differences in bee mortality and *Varroa* control efficacy [68], followed by a Tukey HSD post-hoc test [69].

### Field trials

**Honey bee colonies.**  Three apiaries, separated by a minimum of 5 km, located at The Ohio State University – Wooster campus, were used to conduct the field trial. Each colony consisted of a minimum of two deep 8-frame Langstroth boxes at the start of the experiment. In year 1, each colony was fitted with a screened bottom board for collecting mite drop data. In year 1, 9 hives from apiary 1, and 6 hives each from apiaries 2 and 3 (21 total) were randomly assigned treatments. In year 2, 9 hives from each of the 3 apiaries (27 total) were randomly assigned treatments. In both years, treatments were stratified based on pre-treatment alcohol wash *Varroa* levels and were assigned so that each apiary had an equal number of replicates of each treatment.

**Experimental design.**  Before and after treatment, colonies were assessed by performing *Varroa* alcohol washes and seam counts [70–72]. Seam counts are used to estimate the number of adult bees in a colony and involve two observers visually estimating the clusters or "seams" of bees found between the frames in each box of the hive. Seams in medium boxes were multiplied by the height of a medium Langstroth box divided by the height of a deep Langstroth box (6.625/9.625). Any colonies with less than 9 seams prior to treatment or less than 2 seams after the treatment were excluded from analysis. Alcohol washes were performed by collecting approximately 300 bees from 3 worker brood frames into 70% ethanol, followed by shaking 30 minutes, and counting both bees and *Varroa* that were strained from the wash.

For year 1, *Varroa* that fell below the colony during the treatment, or mite drop, was monitored at 48-hour intervals starting 2 days prior to treatment, at the time of treatment application (day 0), and then 2, 4, 7, 14, and 21 days following treatment. Mite drop was monitored by placing open letter-sized (43.2 cm x 27.9 cm) manila file folders (Staples, Framingham, MA, USA) coated with Vaseline® petroleum jelly (Unilever, Egewood Cliffs, NJ, USA) under screened bottom boards. After folders were removed, they were folded closed and frozen at −20° C for a minimum of 24 hours and stored under these conditions until counted.

**Treatments.**  The combination of oxalic acid active ingredient with Ecostep BC-12® adjuvant was chosen for field trials as it had the highest efficacy in cage trials. In year 1, the adjuvant combination treatment consisted of 1% w/w Ecostep BC-12 adjuvant and 40% w/w oxalic acid dihydrate dissolved in glycerin, the oxalic acid control consisted of 40% w/w oxalic acid dihydrate dissolved in glycerin, and the glycerin control consisted of glycerin only. In year 2, a 0.5% w/w adjuvant concentration was used instead of the 1% concentration in the oxalic acid plus adjuvant treatment. For each uncut Swedish sponge, 70 g of solution, the amount needed for complete saturation of one sponge, was prepared and heated at less than 67° C for at least 1 hour with sonication (Kendal Digital Ultrasonic Heated Cleaner model HB-S-49DHT) and stirring, if necessary.

In year 1, one treatment strip, consisting of a full Swedish sponge (20.3 x 17.8 cm) saturated in treatment solution, was applied for every 9 seams of bees, rounded up, so that each colony had either 2 or 3 treatment strips. In year 2, Swedish

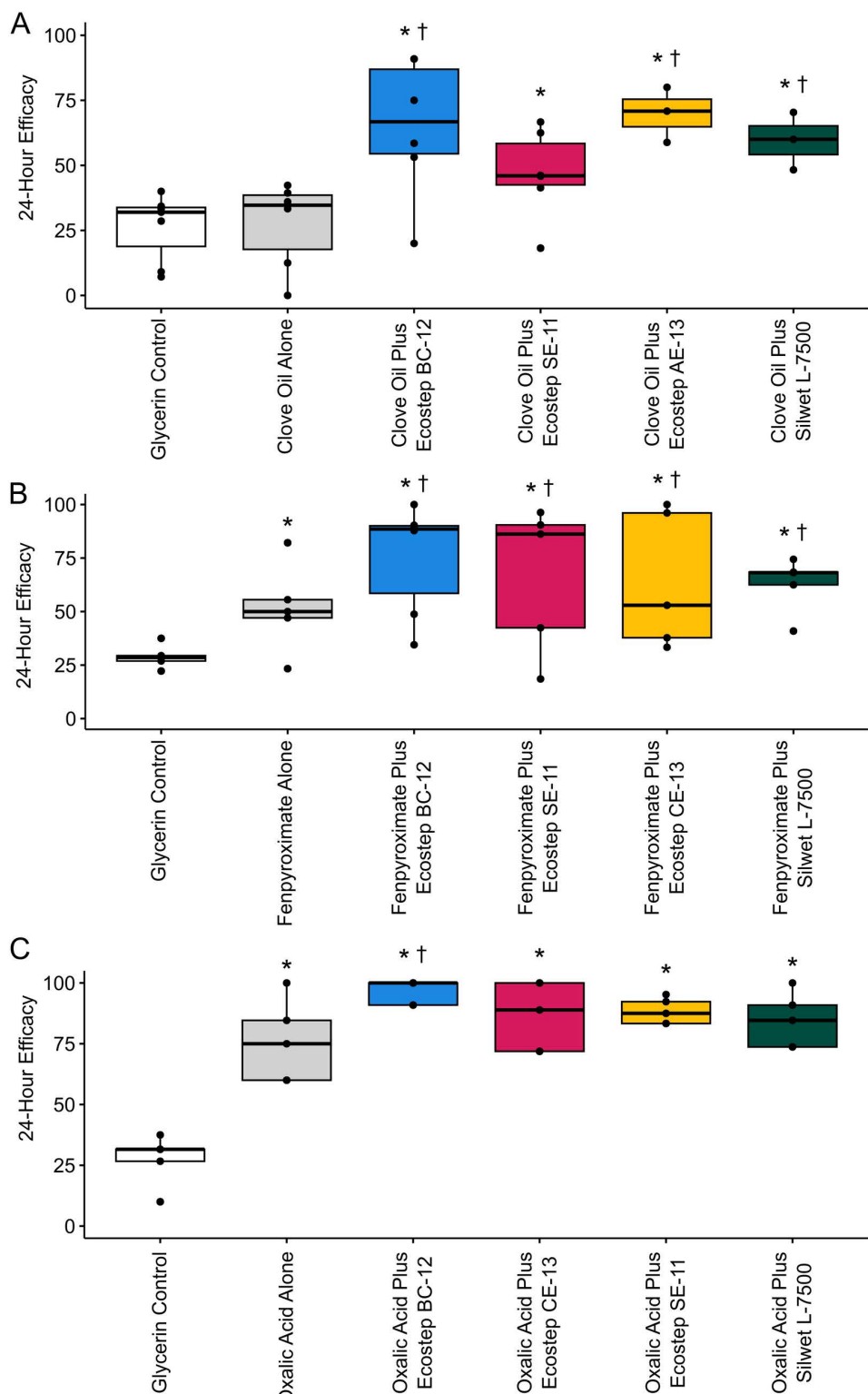

**Fig 1. Efficacy of 3% Clove Oil (A), 0.2% Fenpyroximate (B), and 20% Oxalic Acid (C) treatments in cage trials,** where efficacy is determined to be the number of Varroa mites that fell in 24 hours divided by the total number of Varroa mites in each cage. Adjuvant concentrations were 0.5% for all Ecostep adjuvants and 0.1% for Silwet L-7500 copolymer. Significant difference (P < 0.05) from the glycerin control is indicated by an asterisk (*) and significant difference (P < 0.05) from the active ingredient alone is indicated by a dagger (†).

**Table 2. 48-hour LC$_{50}$ estimates of each adjuvant applied to adult bees using a Potter Spray Tower. All adjuvant LC$_{50}$ estimates were above the maximum rate tested. Values in parenthesis indicate 95% confidence intervals.**

| Adjuvant | Concentrations tested (% v/v) | 48-hour LC50 (% v/v) | Slope (–b) |
|---|---|---|---|
| Ecostep AE-13° | 1, 3, 5, 10, 20 | > 20 | 0.46 (0.16, 0.76) |
| Ecostep BC-12° | 1, 3, 5, 10, 20 | > 20 | 1.07 (0.71, 1.42) |
| Ecostep CE-13° | 1, 3, 5, 10, 20 | > 20 | 0.40 (0.15, 0.66) |
| Ecostep SE-11° | 1, 3, 5, 10, 20 | > 20 | 0.93 (0.60, 1.25) |
| Silwet L-7500 Copolymer° | 0.3, 1, 3, 5, 10, 20 | > 20 | 0.65 (0.25, 1.04) |
| Zeta-cypermethrin Positive Control | 1.70e-4, 5.10e-4, 1.70e-3, 5.10e-3, 1.70e-2 | 0.0039 (0.0035–0.0044) | 2.0 (1.7, 2.4) |

sponges were cut in half to increase contact area with bees, and strips were applied for every 5 seams of bees, rounded up, so that each colony had at least 3 treatment strips. Treatment strips were placed between boxes containing brood. Colonies were left undisturbed during the treatment period. In year 1, treatments were in place for 23 days starting on Sept 21, 2023 for apiary 1, Oct 9 for apiary 2, and Oct 14 for apiary 3. In year 2, treatments were applied for 22 days starting on July 30, 2024 for all apiaries.

**Data analysis.** Field trial results were analyzed independently for each year using the same statistical analysis methods. Efficacy within treatments was compared by modelling individual wash counts using a generalized linear mixed model (GLMM) with a negative binomial distribution and a random intercept for colony [73]. To analyze cumulative mite drop over the full 23-day treatment period in year 1, we modeled the data using a negative binomial generalized linear model [74] to account for overdispersion in count data (S2 Text). The model included treatment and used forward selection to choose variables for inclusion in the model, which included the following predictors: apiary, initial seam count, initial mite wash, and baseline mite drop. A second negative binomial generalized linear model with the same predictors was used for analyzing cumulative mite drop counts over the first 2 and 4 days of treatment.

## Results

### Screening adjuvant toxicity to bees through a spray application

The 48-hour LC$_{50}$ of all adjuvants tested was predicted to be greater than the maximum concentration sprayed with the Potter Tower (Table 2; S3 Dataset), and the mean mortality at the highest tested concentration for each adjuvant was below 50%. The LC$_{50}$ for Mustang Maxx°, the positive control, was estimated to be 0.043%, (95% CI = 0.038, 0.047) when expressed as a formulation concentration, or 0.0039% (95% CI = 0.0035, 0.0044) when expressed as a concentration of zeta-cypermethrin active ingredient.

### Cage trials

**Clove oil.** There was a significant difference in 24-hour bee mortality among treatments (logistic regression, P = 0.028; S1 Text, Table A; S4 Dataset). Tukey's post-hoc test determined that Ecostep BC-12 with clove oil caused significantly more bee mortality than the glycerin control (P = 0.050). A significant difference was observed in 24-hour *Varroa* efficacy (logistic regression, P < 0.001; Fig 1A; S1 Text, Table A; S4 Dataset). Tukey's post-hoc test determined that Ecostep AE-13, Ecostep BC-12, Ecostep SE-11, and Silwet L-7500 Copolymer with clove oil caused significantly increased *Varroa* efficacy (P < 0.05) compared to the glycerin control, and that Ecostep AE-13, Ecostep BC-12, and Silwet L-7500 Copolymer with clove oil caused significantly increased *Varroa* efficacy (P < 0.05) compared to the clove oil alone.

**Fenpyroximate.** There was a significant difference in 24-hour bee mortality among treatments (logistic regression, P < 0.001; S1 Text, Table A; S4 Dataset). Tukey's post-hoc test determined significantly increased bee mortality (P < 0.05)

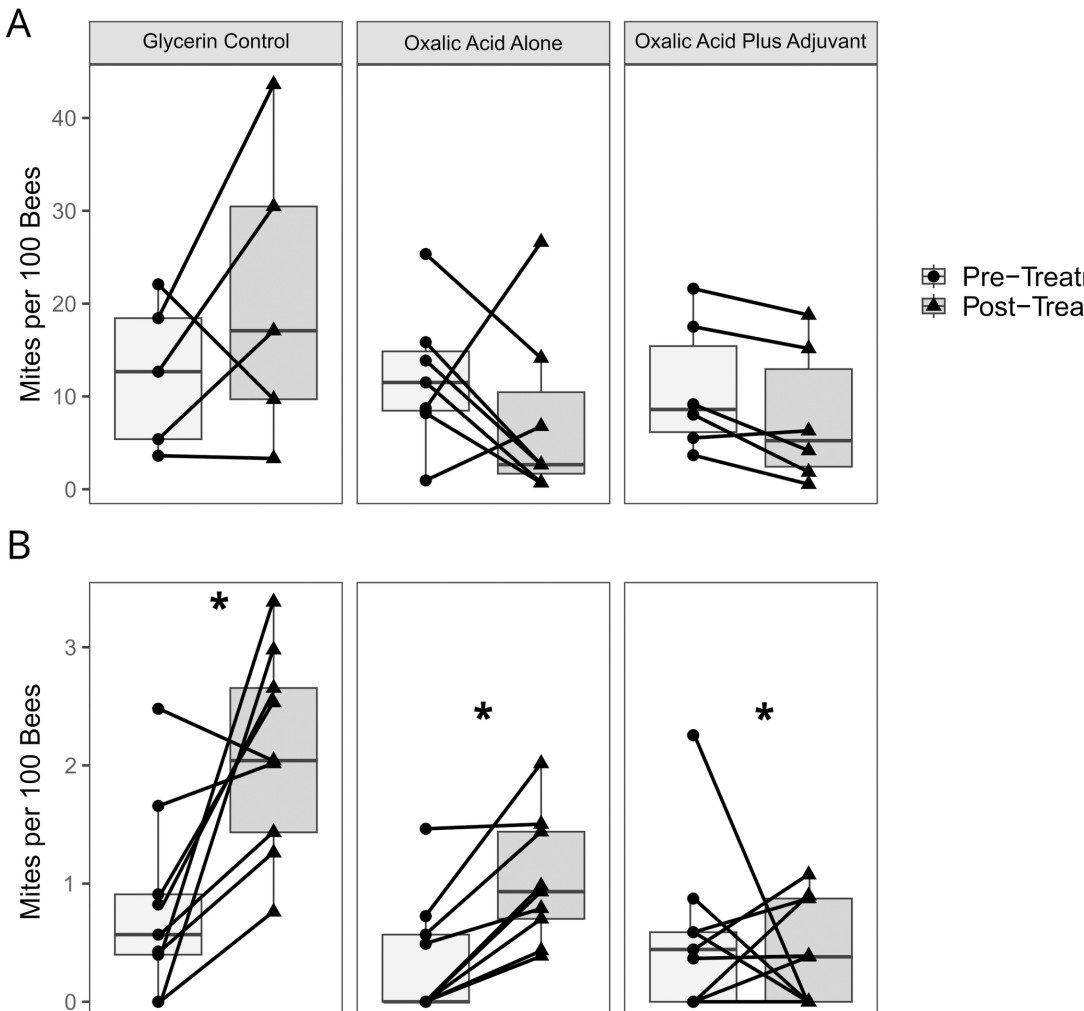

**Fig 2. *Varroa* levels from pre- and post-treatment alcohol washes in year 1 (A) and year 2 (B) field trials.** Points indicate mites per 100 bees for each colony, and lines connect pre- and post-treatment data for each individual colony. A statistically significant difference (P<0.05) between post- and pre-treatment *Varroa* levels within treatments, determined using a GLMM, is indicated with an asterisk (*).

**Table 3. Average daily *Varroa* mite drop over the treatment period. Day 0 is the initial time of treatment. Baseline indicates daily mite drop for the 48-hour period prior to treatment application. Values indicate average daily mite drop during the measurement period for each treatment, with standard deviation listed in parenthesis, where baseline indicates the 48-hour period before treatment.**

| Days after treatment | Baseline | 0-2 | 2-4 | 4-6 | 7-9 | 14-16 | 21-23 |
|---|---|---|---|---|---|---|---|
| Glycerin control | 103 (82.0) | 73.5 (47.1) | 81.9 (48.2) | 72.5 (34.8) | 82.4 (63.5) | 38.9 (24.5) | 38.4 (26.8) |
| Oxalic Acid Control | 109 (72.1) | 171 (99.8) | 377 (160) | 342 (189) | 240 (140) | 104 (65.1) | 66.3 (45.4) |
| Oxalic Acid Plus Adjuvant | 130 (76.8) | 563 (481) | 383 (194) | 288 (195) | 298 (176) | 94.0 (38.7) | 65.1 (47.4) |

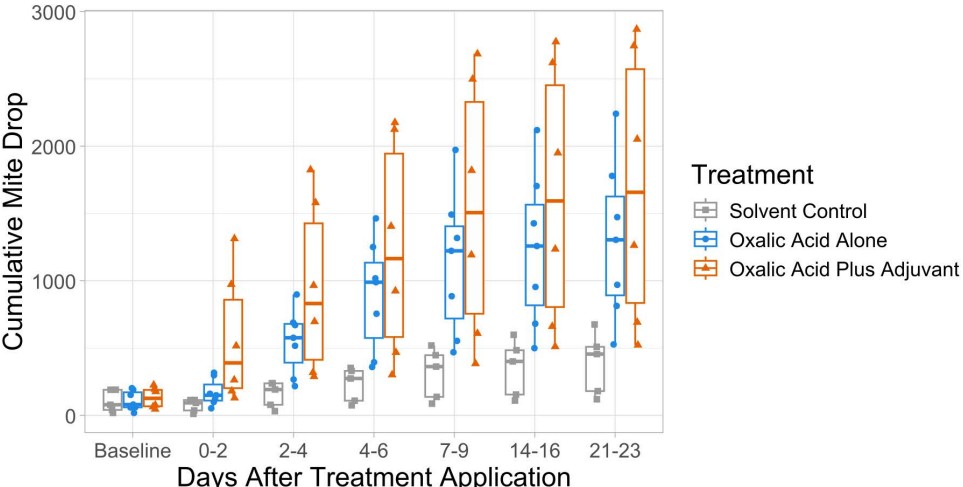

**Fig 3. The cumulative mite drop for colonies in Year 1.** Each point indicates the average daily mite drop for each measurement period plus the sum of all previous measurement periods. The baseline measurement is not included in the cumulative measurement. Treatments were applied at day 0, immediately prior to the start of mite drop sample collection.

for Ecostep AE-13, Ecostep BC-12, and Silwet L-7500 Copolymer with fenpyroximate compared to the glycerin control. A significant difference was observed in 24-hour *Varroa* efficacy (logistic regression, P < 0.001; Fig 1B; S1 Text, Table A; S4 Dataset). Tukey's post-hoc test determined that fenpyroximate alone, as well as Ecostep BC-12, Ecostep CE-13, Ecostep SE-11, and Silwet L-7500 Copolymer with fenpyroximate caused significantly increased *Varroa* efficacy (P < 0.05) compared to the glycerin control, and that Ecostep BC-12, Ecostep CE-13, Ecostep SE-11, and Silwet L-7500 Copolymer with fenpyroximate caused significantly increased *Varroa* efficacy (P < 0.05) compared to the fenpyroximate alone.

**Oxalic acid.** There was a significant difference in 24-hour bee mortality among treatments (logistic regression, P = 0.018; S1 Text, Table A; S4 Dataset). However, Tukey's post-hoc test determined no significant differences in bee mortality between any of the treatments. A significant difference was observed in 24-hour *Varroa* efficacy (logistic regression, P < 0.001; Fig 1C; S1 Text, Table A; S4 Dataset). Tukey's post-hoc test determined that oxalic acid alone, as well as Ecostep BC-12, Ecostep CE-13, Ecostep SE-11, and Silwet L-7500 Copolymer with oxalic acid caused significantly increased *Varroa* efficacy (P < 0.05) compared to the glycerin control, and that only Ecostep BC-12 caused significantly increased *Varroa* efficacy (P < 0.05) compared to the oxalic acid alone. The combination of oxalic acid with Ecostep BC-12 was the most efficacious treatment of all active ingredient-adjuvant combinations, with an efficacy of 96.4% (standard deviation = 5.0).

## Field trials

Colonies that had 2 or fewer seams of bees by the end of the experiment were excluded from analysis. In year 1, the analysis included 5 colonies from the glycerin control group, 7 colonies from the oxalic acid control group, and 6 colonies from the oxalic acid plus adjuvant treatment group. In year 2, no colonies were excluded and all treatment groups included 9 colonies.

**Pre- and post-treatment ethanol washes.** In year 1, the glycerin control colonies had a 1.6-fold increase in *Varroa* per 100 bees (Fig 2A; S1 Text, Table B, Fig B; S5 Dataset), an average increase of 8.4 *Varroa* per 100 bees (95% CI = 1.7, 15.1), from 12.4 (SD = 7.1) to 20.8 (SD = 16.3). However, this increase was not statistically significant (z = 1.203, P > 0.1). The oxalic acid alone had a 43.4% reduction in *Varroa* per 100 bees, which was an average decrease of 4.3 *Varroa* per 100 bees (95% CI = −8.7, 0.1), from 12.1 (SD = 7.6) to 7.6 (SD = 9.6). This decrease was not statistically significant

(z = −1.576, P > 0.1). A 35.2% reduction in *Varroa* per 100 bees was observed between the pre- and post-treatment ethanol washes for the oxalic acid plus adjuvant treatment, which was an average decrease of 3.1 *Varroa* per 100 bees (95% CI = −4.1, −2.1), from 10.9 (SD = 7.1) to 7.8 (SD = 7.5). This decrease was also not statistically significant (z = −1.155, P > 0.1).

In year 2, a 2.6-fold increase in *Varroa* per 100 bees was observed for colonies in the glycerin control (Fig 2B; S1 Text, Table B, Fig B; S5 Dataset), an average increase of 1.3 *Varroa* per 100 bees (95% CI = 0.9, 1.7), from 0.8 (SD = 0.8) to 2.1 (SD = 0.9). This increase was statistically significant (z = 22.020, p < 0.0001). In the oxalic acid alone treatment, *Varroa* per 100 bees increased 2.8-fold after treatment, an average increase of 0.7 *Varroa* per 100 bees (95% CI = 0.5, 0.8), from 0.4 (SD = 0.5) to 1.0 (SD = 0.5). This increase was statistically significant (z = 16.102, p < 0.0001). In the oxalic acid plus adjuvant treatment, a 29% reduction in *Varroa* per 100 bees was observed between the pre- and post-treatment ethanol washes, which was an average decrease of 0.2 *Varroa* per 100 bees (95% CI = −0.5, 0.2), from 0.6 (SD = 0.7) to 0.4 (SD = 0.4). This decrease was statistically significant (z = −5.016, P < 0.0001).

**Varroa mite drop.** Over the 23-day treatment period in year 1, colonies treated with oxalic acid with adjuvant had a 4.7-fold increase in cumulative mite drop compared to the glycerin control (p < 0.001; Table 3; Fig 3; S5 Dataset). In the first 4 days, colonies treated with oxalic acid with adjuvant had a 6.4-fold increase in total mite drop compared to the glycerin control (p < 0.001), and a 1.46-fold increased total mite drop compared to oxalic acid alone, but this difference was not statistically significant (p = 0.117). In the first 2 days, colonies treated with adjuvant had an 8.5-fold increase in mite drop compared to the glycerin control (P < 0.001), and a 2.8-fold increase in mite drop compared to oxalic acid alone, which was statistically significant (P < 0.001).

## Discussion

Ecostep BC-12 adjuvant increased the efficacy of oxalic acid in cage trials without any increase in honey bee mortality, with an average efficacy exceeding 96%. In the field trial in year 1, neither oxalic acid alone nor oxalic acid combined with Ecostep BC-12 resulted in a statistically significant reduction in *Varroa* measured with an alcohol wash. The control group also showed no increase in *Varroa* per 100 bees over the course of the experiment. In the field trial in year 2, the results were more definitive: colonies treated with oxalic acid alone experienced a 184% increase in mite counts following treatment, indicating that oxalic acid without an adjuvant was ineffective under these field conditions. In contrast, colonies treated with oxalic acid combined with Ecostep BC-12 experienced a 29% reduction in mite loads, demonstrating clear efficacy when the adjuvant was included. While oxalic acid extended-release formulations in glycerin have been found to be effective in some studies [75–78], other studies found them to be ineffective [79]. No significant changes in mite levels were observed for any of the three treatments in year 1, when pre-treatment *Varroa* levels were high (average 11.8 mites per 100 bees across all treatment groups). In year 2, when pre-treatment *Varroa* was lower (average 0.58 mites per 100 bees across all treatment groups), oxalic acid alone did not provide effective *Varroa* control, but oxalic acid combined with Ecostep BC-12 did significantly reduce *Varroa* numbers. Mite drop data from year 1 indicated improved efficacy over the full 23-day treatment period for the oxalic acid with adjuvant treatment compared to glycerin control, but no difference between the oxalic acid with adjuvant and oxalic acid alone. However, over the first two days of treatment, increased mite drop was observed for the oxalic acid with adjuvant treatment relative to oxalic acid alone, indicating a more immediate impact of treatment when an adjuvant was included.

The decreased mite drop observed between the collection periods of days 7–9 and days 14–16 may indicate that extended-release oxalic acid treatments may become less effective over time (Table 3; S1 Text, Fig C). While studies have been performed on similar extended-release oxalic acid treatments, no published studies have addressed the persistence of oxalic acid in the treatments beyond taking measurements of *Varroa* levels in colonies. Yet, typical extended-release products for *Varroa* control are labeled for 42–56 days, indicating that residue testing on treatment strips would be useful in determining whether treatments are effective for a full 42- to 56-day treatment period. Comparisons of oxalic acid

movement throughout the colony at multiple time points [80] would also be useful. Further testing is required to determine the optimal oxalic acid and adjuvant concentrations, and whether glycerin alone or a glycerin-water mixture should be used. Other off-label oxalic acid extended-release formulations have used a mixture of glycerin and water [78,79], which may change the activity of an adjuvant. While field experiments were performed in three geographically distinct apiaries, efficacy for many *Varroa* control miticides can vary with climate [79,81–84]. Further testing of this formulation should be performed in other climatic regions.

The increased mite drop observed over the first two days when an adjuvant was included may be the result of increased spreading throughout the colony. As all five adjuvants used in this study are listed by the manufacturer as surfactants, they may improve spreading on the honey bees or the *Varroa* themselves, thereby increasing the exposure of *Varroa* to the miticide. Increased spreading of the miticide throughout the colony and over the bee cuticle is critical in mediating exposure of *Varroa* to the miticide, as most *Varroa* on adult bees are located under the honey bee abdominal sternites [85]. In addition to improving spreading, adjuvants may also act through mechanisms that improve spiracular or cuticular penetration of the miticide in *Varroa*, as demonstrated in other arthropod pests [50,86–89]. It is also possible that adjuvants may have intrinsic toxicity to *Varroa*, as has been observed in spider mites [89–92], aphids [92–95], thrips [96], cockroaches [97], and mosquitoes [43,98]. However, the adjuvants used in this study were not tested on their own for toxicity to *Varroa*. Other adjuvants not used in this study are toxic to honey bees [45,99–104] and other bee species [105,106], so it is important to perform testing prior to including other adjuvants in candidate miticide formulations. The mechanism of action for adjuvant toxicity to arthropods is not well understood, but it is thought that adjuvants act similarly to insecticidal soaps [45,91,107,108], which can disrupt the arthropod cuticle, break down cell membranes, and reduce water surface tension to cause spiracular drowning [98,109].

Five adjuvant products, including two polyethylene/polypropylene glycol ethoxylates (Ecostep AE-13˚ and BC-12˚), two fatty acid ethoxylates (Ecostep CE-13˚ and SE-11˚), and one organo-silicone (Silwet L-7500 Copolymer˚), were investigated in this study, but there are many other types of adjuvants that may improve efficacy of *Varroa* control applications, including other non-ionic surfactants, crop oils, seed oils, hydrocolloid polymers, or combinations [110]. Formulations of *Varroa* control products could take advantage of the spectrum of adjuvants to enhance new active ingredients for *Varroa* control [40,111]. Oxalic acid extended-release products utilizing adjuvants should be further developed and registered for use through the US EPA or the appropriate regulatory agency prior to use by beekeepers. There is a need to improve strategies for managing *Varroa*. Improving formulation chemistry is a critical tool to improve efficacy of chemical miticides that can be incorporated into an integrated pest management program [12,112]. This study is the first to demonstrate that adjuvants can improve the efficacy of miticide active ingredients against *Varroa*, which will provide better tools for beekeepers to control this devastating pest.

## Supporting information

**S1 Text. Supplemental Tables and Figures. S1 Table A.** Bee mortality and *Varroa* mite drop efficacy for cage trials, expressed as a percentage, with range of standard error listed in parenthesis. The P-value of each treatment compared to the glycerin control and active ingredient control were determined via a pairwise Wilcoxon Rank-Sum Test using Benjamini-Hochberg post-hoc correction. Statistically significant differences ($P < 0.05$) are indicated with an asterisk (*).**S1 Table B.** Year 1 and 2 ethanol wash data for each treatment, where rate is defined as *Varroa* infestation rate per 100 bees. The value listed in parenthesis for the initial and final rate mean indicates standard deviation; the range listed in parenthesis for the mean difference indicates standard error. The column "n" indicates the number of colonies used for analysis for each treatment. P-value within treatments was determined using a one-sided t-test on the difference within treatments and P-value between treatments and each of the controls was determined with an ANOVA with Tukey's Post-Hoc test of the change in mite levels, where an asterisk (*) indicates a statistically significant test ($P < 0.05$).**S1 Fig A.** Cage design for laboratory cage trials, with approximately 300 bees added to each cage.**S1 Fig B.** The change in *Varroa*

levels from alcohol washes for year 1 (A) and year 2 (B) field trials. Points indicate the post-treatment minus the pre-treatment *Varroa* mites per 100 bees for individual colonies. A statistically significant difference (P<0.05) between post- and pre-treatment *Varroa* levels within treatments, determined using a GLMM, is indicated with an asterisk (*).**S1 Fig C.** The daily mite drop for colonies in Year 1. Each point indicates the average daily mite drop over the 48-hour measurement period for an individual colony. Treatments were applied at day 0, immediately prior to sample collection. (DOCX)

**S2 Text. R Code for Cage and Field Trial Data Analysis.**
(DOCX)

**S3 Dataset. Adjuvant Toxicity Testing Dataset.** Adjuvant dose is listed as a percent composition by volume. Insecticide dose is listed as multiples of the application rate (X), where 1X Mustang Maxx is equivalent to 0.0186 percent composition by volume of formulation, or 0.00170 percent composition by weight of zeta-cypermethrin active ingredient. (XLSX)

**S4 Dataset. Cage Trial Dataset.**
(XLSX)

**S5 Dataset. Year 1 Field Trial Dataset.**
(XLSX)

**S6 Dataset. Year 2 Field Trial Dataset.**
(XLSX)

## Acknowledgments

We thank Brooke Fries and Emily Greenland for their contribution to methods development. We thank Lauren Tarver for her contributions to acute toxicity testing, cage manufacturing, setup of laboratory assays, collection of sticky boards, and sticky board *Varroa* counts. We thank Makayla Phillips for her contributions to setup of laboratory assays, setup of field trials, collection of sticky boards, and counting *Varroa* in sticky boards and ethanol washes. We thank Frank Rinkevich for advice on cage test methodology. We thank Stepan for providing the Ecostep adjuvant samples and Momentive for providing the Silwet adjuvant sample.

## Author contributions

**Conceptualization:** Brandon Shannon, Reed M. Johnson.

**Data curation:** Brandon Shannon, Lucas Marsh.

**Formal analysis:** Brandon Shannon, Rui Zhang, Reed M. Johnson.

**Funding acquisition:** Brandon Shannon, Reed M. Johnson.

**Investigation:** Brandon Shannon, Lucas Marsh, Reed M. Johnson.

**Methodology:** Brandon Shannon, Rui Zhang, Lucas Marsh, Reed M. Johnson.

**Project administration:** Brandon Shannon, Reed M. Johnson.

**Resources:** Reed M. Johnson.

**Supervision:** Reed M. Johnson.

**Validation:** Rui Zhang, Reed M. Johnson.

**Visualization:** Brandon Shannon, Reed M. Johnson.

Writing – original draft: Brandon Shannon.

Writing – review & editing: Brandon Shannon, Rui Zhang, Lucas Marsh, Reed M. Johnson.

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
