## [Decision Letter · Decision Letter 0]

23 Mar 2025

Dear Dr. Shannon,

Thank you for submitting your manuscript to PLOS ONE. After careful consideration, we feel that it has merit but does not fully meet PLOS ONE’s publication criteria as it currently stands. Therefore, we invite you to submit a revised version of the manuscript that addresses the points raised during the review process.

Specifically, there are some concerns about the presentation expressed by both reviewers and more importantly, the second reviewer raises some concerns about the statistical models you used. Please either justify what you did or revise as suggested.

We look forward to receiving your revised manuscript.

Kind regards,

Olav Rueppell

Academic Editor

PLOS ONE

Journal Requirements:

We thank the National Honey Board with Project Apis m., the California State Beekeeper’s Association, The Ohio State University CFAES Internal Grants Program, USDA-NIFA-SCRI (2023-51181-41246) and state and federal appropriations to The Ohio State University College of Food, Agriculture and Environmental Science (OHO01558-MRF). 

I have read the journal's policy and the authors of this manuscript have the following competing interests: A patent application (PCT/US2024/056156; ADJUVANTS TO IMPROVE EFFICACY OF VARROA CONTROL ACTIVE INGREDIENTS IN MANAGED HONEY BEE COLONIES) was filed on November 15, 2024, by the Ohio State Innovation Foundation with inventors Reed Johnson and Brandon Shannon.

5. Please provide a complete Data Availability Statement in the submission form, ensuring you include all necessary access information or a reason for why you are unable to make your data freely accessible. If your research concerns only data provided within your submission, please write "All data are in the manuscript and/or supporting information files" as your Data Availability Statement.

6. We are unable to open your Supporting Information file [S2_File.R]. Please kindly revise as necessary and re-upload.

Reviewers' comments:

Reviewer's Responses to Questions

**Comments to the Author**

1. Is the manuscript technically sound, and do the data support the conclusions?

Reviewer #1: Yes

Reviewer #2: Partly

2. Has the statistical analysis been performed appropriately and rigorously?

Reviewer #1: I Don't Know

Reviewer #2: No

3. Have the authors made all data underlying the findings in their manuscript fully available?

Reviewer #1: Yes

Reviewer #2: Yes

4. Is the manuscript presented in an intelligible fashion and written in standard English?

Reviewer #1: Yes

Reviewer #2: Yes

Reviewer #1: General comments:

This study by Shannon et al. investigated the efficacy of oxalic acid and other acaricides in glycerin formulations combined with several adjuvants for the control of the parasitic mite Varroa destructor. A question of interest was if adjuvants increase the varroacidal efficacy of acaricides. Laboratory cage assays were used to screen the acaricidal effect of three products in combination with several adjuvants. One acaricide (oxalic acid) and one adjuvant (Ecostep BC-12) were selected for field testing. Results indicate that the adjuvant increased the efficacy of oxalic acid against Varroa, but the acaricide was ineffective relative to the control, when used alone. The manuscript is well written and in my opinion only requires a few minor editions and reorganization for improvement that I mention under specific comments.

Specific comments:

1. Titles of sections (Abstract, Methods, Results, etc.) are typed in red font. Is this a requirement of the journal? If so, leave it as it is, if not, please follow the instructions for authors of the journal.

2. Abstract, L23. “soaked strips; field trials evaluated oxalic acid combined with..” I suggest using a period instead of a semicolon to separate lab trials from field trials. Also, please mention why oxalic acid and Ecostep BC-12 were selected for field trials.

3. Abstract. Please provide some numerical and statistical comparisons from your results between the treatment with the adjuvant and the treatment without the adjuvant.

4. Methods, Treatments, L216-28. Please mention somewhere in Methods why oxalic acid and Ecostep BC-12 were selected for field trials.

5. Data Analysis L233-239. Differences in Varroa levels were either analyzed with t tests or ANOVAs. Were these data checked for normality and homoscedasticity before analyzing them? If so, what tests did you use? Please mention them.

6. Discussion. I suggest you to start with a small paragraph outlining your main results, both, from cages and from hives. Following paragraphs should discuss your results and conclusions in comparison with other studies previously published.

7. Discussion first and second paragraphs (L 330-349) are justifications of methods used. I suggest shortening the text containing this information and move the text to the Methods section.

8. Discussion L 371-374. Again, the arguments presented here are a justification to conduct the study and thus they belong in the Introduction section.

9. Conclusion. I suggest first mentioning results and conclusions from your cage experiments before mentioning those of your field trial experiments.

10. Conclusion L 406: “oxalic alone” Do you mean “oxalic acid alone”?

Reviewer #2: This seems to be a well-conducted study. Appropriate methods are used for the LC50 analysis. A few more key details are needed: sample size for cage trials (number of cages per treatment), and initial and ending mite levels for the field experiment.

I have some suggestions (in blue text in the attachment) for alternative statistical approaches for some the analyses, mostly to incorporate the “generalized” version of the linear modeling. I think reanalyses will let you say more about the effect of adjuvants in cage trials, but likely you will need to pull back some of your statements about significant effects in the field trial.

I appreciate you including your data, as this allowed me to test some alternative approaches. At the end of this document I have included the R code I used when reviewing your analyses. If you paste this into an R script, some of this code may be helpful, or at least will let you follow my process of reviewing your stats.

**Do you want your identity to be public for this peer review?** For information about this choice, including consent withdrawal, please see our Privacy Policy

Reviewer #1: No

Reviewer #2: No

---

## [Author Response · Author response to Decision Letter 1]

7 May 2025

Thank you to the editor and reviewers for taking the time to review our manuscript and provide valuable feedback. We have modified the Funding Information, Competing Interested Statement, and Data Availability Statement to match the requirements of PLOS ONE. We have made modifications to our data analysis in response to Reviewer 2’s concerns. Please see responses to specific comments in the attached "Response to Reviewers" document or in the text below.

Journal Requirements:

Thank you, we have ensured that our file names match the requirements for PLOS ONE.

Thank you, we have ensured that the grant numbers in the ‘Funding Information’ section of the manuscript submission are correct.

We thank the National Honey Board with Project Apis m., the California State Beekeeper’s Association, The Ohio State University CFAES Internal Grants Program, USDA-NIFA-SCRI (2023-51181-41246) and state and federal appropriations to The Ohio State University College of Food, Agriculture and Environmental Science (OHO01558-MRF).

Thank you, we have added this statement to the manuscript and included the full revised statement in the Cover Letter.

I have read the journal's policy and the authors of this manuscript have the following competing interests: A patent application (PCT/US2024/056156; ADJUVANTS TO IMPROVE EFFICACY OF VARROA CONTROL ACTIVE INGREDIENTS IN MANAGED HONEY BEE COLONIES) was filed on November 15, 2024, by the Ohio State Innovation Foundation with inventors Reed Johnson and Brandon Shannon.

Thank you, we have added this statement to the manuscript and included the full revised statement in the Cover Letter.

5. Please provide a complete Data Availability Statement in the submission form, ensuring you include all necessary access information or a reason for why you are unable to make your data freely accessible. If your research concerns only data provided within your submission, please write "All data are in the manuscript and/or supporting information files" as your Data Availability Statement.

Thank you, we have added a Data Availability Statement in the manuscript submission: “All data can be found in the manuscript and/or supporting information files. Code used for statistical analysis can be found in the supporting information files.”

6. We are unable to open your Supporting Information file [S2_File.R]. Please kindly revise as necessary and re-upload.

Thank you, we have revised S2 as a text file instead of an R script file.

Thank you, we have reviewed our references. They are correct to the best of our knowledge.

Reviewers' comments:

Reviewer's Responses to Questions

Comments to the Author

1. Is the manuscript technically sound, and do the data support the conclusions?

Reviewer #1: Yes

Reviewer #2: Partly

2. Has the statistical analysis been performed appropriately and rigorously?

Reviewer #1: I Don't Know

Reviewer #2: No

3. Have the authors made all data underlying the findings in their manuscript fully available?

Reviewer #1: Yes

Reviewer #2: Yes

4. Is the manuscript presented in an intelligible fashion and written in standard English?

Reviewer #1: Yes

Reviewer #2: Yes

5. Review Comments to the Author

Reviewer #1: General comments:

This study by Shannon et al. investigated the efficacy of oxalic acid and other acaricides in glycerin formulations combined with several adjuvants for the control of the parasitic mite Varroa destructor. A question of interest was if adjuvants increase the varroacidal efficacy of acaricides. Laboratory cage assays were used to screen the acaricidal effect of three products in combination with several adjuvants. One acaricide (oxalic acid) and one adjuvant (Ecostep BC-12) were selected for field testing. Results indicate that the adjuvant increased the efficacy of oxalic acid against Varroa, but the acaricide was ineffective relative to the control, when used alone. The manuscript is well written and in my opinion only requires a few minor editions and reorganization for improvement that I mention under specific comments.

Thank you to Reviewer 1 for taking the time to review our manuscript and provide constructive feedback.

Specific comments:

1. Titles of sections (Abstract, Methods, Results, etc.) are typed in red font. Is this a requirement of the journal? If so, leave it as it is, if not, please follow the instructions for authors of the journal.

Font color has been changed to black to match journal formatting guidelines.

2. Abstract, L23. “soaked strips; field trials evaluated oxalic acid combined with..” I suggest using a period instead of a semicolon to separate lab trials from field trials. Also, please mention why oxalic acid and Ecostep BC-12 were selected for field trials.

Thank you, this has been revised.

“in glycerin-soaked strips. Field trials evaluated the best performing active ingredient-adjuvant combination from cage trials, oxalic acid combined with Ecostep BC-12® adjuvant in glycerin-soaked strips.”

3. Abstract. Please provide some numerical and statistical comparisons from your results between the treatment with the adjuvant and the treatment without the adjuvant.

Please see the following revisions:

“Field trials with oxalic acid and adjuvant in glycerin caused a significant decrease of 3.1 (SE: 2.1 – 4.1) Varroa per 100 bees in year 1 and a significant reduction in Varroa, relative to the solvent control, from 2.1 to 0.4 mites per 100 bees, in year 2.”

4. Methods, Treatments, L216-28. Please mention somewhere in Methods why oxalic acid and Ecostep BC-12 were selected for field trials.

Please see the following revisions to the methods section:

L226: “The combination of oxalic acid active ingredient with Ecostep BC-12® adjuvant was chosen for field trials as it had the highest efficacy in cage trials.”

5. Data Analysis L233-239. Differences in Varroa levels were either analyzed with t tests or ANOVAs. Were these data checked for normality and homoscedasticity before analyzing them? If so, what tests did you use? Please mention them.

Thank you for pointing this out. Our statistical methods have been changed to using a generalized linear mixed model (GLMM) to determine pre- and post-treatment differences. Please see text excerpt for description of revised statistical methods:

L246: “Efficacy within treatments was compared by modelling individual wash counts using a generalized linear mixed model (GLMM) with a negative binomial distribution and a random intercept for colony (73).”

6. Discussion. I suggest you to start with a small paragraph outlining your main results, both, from cages and from hives. Following paragraphs should discuss your results and conclusions in comparison with other studies previously published.

Thank you, we have moved the start of the conclusion that describes the main results to the start of the discussion.

7. Discussion first and second paragraphs (L 330-349) are justifications of methods used. I suggest shortening the text containing this information and move the text to the Methods section.

Thank you, this has been shortened and moved to the methods section.

8. Discussion L 371-374. Again, the arguments presented here are a justification to conduct the study and thus they belong in the Introduction section.

Thank you, this has been moved to the introduction.

9. Conclusion. I suggest first mentioning results and conclusions from your cage experiments before mentioning those of your field trial experiments.

This is a great suggestion, we have moved the first half of the conclusion paragraph to the beginning of the discussion and removed the section header “Conclusion”.

10. Conclusion L 406: “oxalic alone” Do you mean “oxalic acid alone”?

Thank you, this has been corrected.

Reviewer #2:

Overall

This seems to be a well-conducted study. Appropriate methods are used for the LC50 analysis. A few more key details are needed: sample size for cage trials (number of cages per treatment), and initial and ending mite levels for the field experiment.

I have some suggestions (in blue text) for alternative statistical approaches for some the analyses, mostly to incorporate the “generalized” version of the linear modeling. I think reanalyses will let you say more about the effect of adjuvants in cage trials, but likely you will need to pull back some of your statements about significant effects in the field trial.

I appreciate you including your data, as this allowed me to test some alternative approaches. At the end of this document I have included the R code I used when reviewing your analyses. If you paste this into an R script, some of this code may be helpful, or at least will let you follow my process of reviewing your stats.

We thank Reviewer 2 for taking the time to review our manuscript and provide constructive feedback. We are grateful for your contributions and suggestions for the statistical analysis.

Abstract

- Nice and clear

Introduction

- L38-39. Remove the word “average” to reflect the kind of estimate that’s made, an estimate of the loss rate for the population of honey bee colonies. For the CAPA survey please confirm, but you will probably remove “average” here as well.

Thank you for catching this, “average” has been removed for reference to both surveys.

- L41-42. This is a little different than what you have written. 60% of commercial beekeeping operations reported it as a leading cause of winter loss.

Thank you, this has been corrected.

- L42-43. This list of pathogens may not be necessary since “associated diseases” was already mentioned.

This sentence has been removed.

- L67. Spelling of Mitchondrial

Thank you, this has been corrected.

- L84. Reconsider word choice of “greaten”

Changed to “increase”

- L93. Capitalization and italicization of “Varroa”

Thank you, this has been corrected.

Materials and Methods

- Data analysis methods for the LC50 determination should be briefly added. Reading the manuscript I think you used the drc package to fit 2-parameter log-logistic dose-response models.

Please see the following added sentence describing data analysis:

L123: “Data analysis was performed with the drc package in R (52) using 2-parameter log-logistic dose-response models, as performed by (45).”

45. Shannon B, Walker E, Johnson RM. Toxicity of spray adjuvants and tank mix combinations used in almond orchards to adult honey bees (Apis mellifera). J Econ Entomol. 2023 Oct 1;116(5):1467–80.

52. Ritz C, Baty F, Streibig JC, Gerhard D. Dose-Response Analysis Using R. PLOS ONE. 2015 Dec 30;10(12):e0146021.

- L97. Do you have an analysis of the chemical constituents of the clove oil used? Or if not, any reference for usual composition and / or how much these vary?

- L99. How is 102.5% purity possible? Spelling of Scientific

Spelling corrected. 102.5% purity is not an error; this is the specification from Thermo Fisher Scientific. It is likely that purity above 100% is possible because oxalic acid that is not fully hydrated would have a greater relative concentration of oxalic acid compared to pure oxalic acid dihydrate. I have listed below the COA for our most recent lot of Oxalic Acid Dihydrate, which has specifications for 99.5 – 102.5%.

https://www.fishersci.com/us/en/catalog/search/cofa/getcert?catalognumber=A219&lotnumber=242439&docType=01

- L127 and L134-136. Only one of these statements is needed.

Thank you for catching this repeated statement, the statement at line 127 has been removed.

- L133. “Sonicating” or rephrase

Corrected

- L152. Were the weigh boats or trays under the cages covered with Vaseline as in the Rinkevich protocol? I am only asking since it is not mentioned here or on L180

We had only mentioned deviations from the Rinkevich protocol. We added the following sentence to reduce confusion.

L160: “The design of cages above petroleum jelly-coated weigh boats was modified from Rinkevich, 2020 (17) (S1 Text, Fig A).”

- L170. Replace “phoretic mites” with “mites in the dispersal phase” or “mites on adult bees” as phoresy refers to non-feeding transport.

Thank you, this has been correcte

---

## [Editor Report · Decision Letter 1]

13 May 2025

Adjuvants to improve efficacy of miticides in managed honey bee (Apis mellifera) colonies to control Varroa destructor

PONE-D-25-07730R1

Dear Dr. Shannon,

We’re pleased to inform you that your manuscript has been judged scientifically suitable for publication and will be formally accepted for publication once it meets all outstanding technical requirements.

Kind regards,

Olav Rueppell

Academic Editor

PLOS ONE
---

## [Editor Report · Acceptance letter]

PONE-D-25-07730R1

PLOS ONE

Dear Dr. Shannon,

I'm pleased to inform you that your manuscript has been deemed suitable for publication in PLOS ONE. Congratulations! Your manuscript is now being handed over to our production team.

Kind regards,

on behalf of

Dr. Olav Rueppell

Academic Editor

PLOS ONE